# Streamlining YOLOv7 for Rapid and Accurate Detection of Rapeseed Varieties on Embedded Device

**DOI:** 10.3390/s24175585

**Published:** 2024-08-28

**Authors:** Siqi Gu, Wei Meng, Guodong Sun

**Affiliations:** 1School of Information Science and Technology, Beijing Forestry University, Beijing 100083, China; aguuu_547@bjfu.edu.cn (S.G.); sungd@bjfu.edu.cn (G.S.); 2Engineering Research Center for Forestry-Oriented Intelligent Information Processing, Beijing 100083, China

**Keywords:** rapeseed detection, embedded device, pruning strategy, inference speed

## Abstract

Real-time seed detection on resource-constrained embedded devices is essential for the agriculture industry and crop yield. However, traditional seed variety detection methods either suffer from low accuracy or cannot directly run on embedded devices with desirable real-time performance. In this paper, we focus on the detection of rapeseed varieties and design a dual-dimensional (spatial and channel) pruning method to lighten the YOLOv7 (a popular object detection model based on deep learning). We design experiments to prove the effectiveness of the spatial dimension pruning strategy. And after evaluating three different channel pruning methods, we select the custom ratio layer-by-layer pruning, which offers the best performance for the model. The results show that using custom ratio layer-by-layer pruning can achieve the best model performance. Compared to the YOLOv7 model, this approach results in mAP increasing from 96.68% to 96.89%, the number of parameters reducing from 36.5 M to 9.19 M, and the inference time per image on the Raspberry Pi 4B reducing from 4.48 s to 1.18 s. Overall, our model is suitable for deployment on embedded devices and can perform real-time detection tasks accurately and efficiently in various application scenarios.

## 1. Introduction

Rapeseed (*Brassica napus* L.) is an important crop widely planted around the world, mainly used for producing animal feed, extracting vegetable fats, and as a raw material for biodiesel. According to FAOSTAT (https://www.fao.org/faostat/zh/#data, accessed on 1 July 2024) data, the global area planted with rapeseed has reached 39.97 million hectares with a production of approximately 87.22 million tons. China is the second-largest producer of rapeseed. As of July 2021, China has registered 1212 rapeseed varieties. Different varieties of rapeseed display distinct phenotypic characteristics and yield differences. The confusion of different varieties of seeds, which can occur during processing, planting, handling, and storage, could lead to uneven growth, heighten susceptibility to pests and diseases, and ultimately damage the rapeseed quality. As economic well-being increases, the demand for rapeseed oil continues to grow, along with heightened expectations for both product quality and diversity. Therefore, it has become an important issue to effectively distinguish rapeseed varieties.

In conventional studies, researchers have identified rapeseed varieties based on phenotypic characteristics [1,2,3]. However, these methods rely on experts to identify each seed individually. This process is not only time-consuming but also requires substantial labor, thereby limiting its overall efficiency. Additionally, the volume of rapeseed is usually small, resulting in a low accuracy rate when identified using naked eyes. More recently, machine learning [4] has been applied to identifying seed varieties [5,6,7]. Although the application of machine learning marks an advancement in rapeseed detection, it still has several drawbacks. Machine learning methods rely on manual feature engineering, which depends heavily on substantial professional knowledge and experience. Furthermore, the constructed characteristics exhibit challenges in their adaptability to diverse scenarios and datasets, consequently constraining the model’s capacity for generalization.

Deep learning, a sub-field of machine learning, excels in automatically extracting features by a multi-layered deep neural network (DNN, for short) [8]. DNNs are well suited for learning from data with intricate structures. In agriculture, the use of DNNs for crop identification and object detection is gaining significance, with works [9,10,11] demonstrating the effectiveness of DNNs in detecting various types of crop seeds. Specifically, DNN excels not only in identifying seed variety but also in pinpointing seeds as objects without feature engineering, which are harnessed to present results to users in a way that is both more efficient and user-friendly. Such strengths are showcased by several recent studies. To detect damage to cotton seeds, Liu et al. [12] use the YOLO (a popular DNN backbone for object detection) and integrate a lightweight upsampling operator to improve the accuracy. Their experimental results show a mAP (mean Average Precision) of 99.5% and a recall rate of 99.3%. In the experiment for leguminous seed variety identification [13], YOLOv4 [14] outperforms Faster R-CNN [15] in terms of accuracy and detection capability, offering an effective method for detecting different legume seeds under complex conditions. Ng et al. [16] use DNN to simultaneously process a large number of oil palm seeds based on a set of multi-view images, which reduces the effort in data collection. Their oil palm seed quality DNN model can achieve an accuracy rate of 90% on multi-view images.

YOLOv7 is a popular deep neural network model used for object detection tasks. YOLOv7 is known for its balance between speed and accuracy, making it suitable for a wide range of real-time applications. In addition to the standard YOLOv7 model, there are variants such as YOLOv7x and YOLOv7-tiny. YOLOv7x is an extended version that offers higher accuracy and better performance for more complex tasks, albeit at the cost of increased computational resources. On the other hand, YOLOv7-tiny is a smaller, faster version designed for deployment in resource-constrained environments, making it ideal for applications requiring rapid inference with limited hardware capabilities. YOLOv7 and its variants find extensive applications in agriculture, including pest and disease detection [17,18], fruit classification [19,20], and weed management [21,22]. The flexibility and efficiency of these models make them invaluable tools for enhancing precision agriculture and improving crop yields.

In the field of seed detection applications, object detection models exhibit two main limitations. First, most models cannot meet the reliability requirements of real-world applications for accuracy. Second, in terms of adaptability, most models struggle to operate efficiently in real-world applications, especially in scenarios with limited computational resources. Therefore, designing accurate and practical models is a significant issue in the community. To address these limitations, model compression strategies have proven effective in resource-constrained scenarios such as autonomous driving [23], security surveillance [24], and e-commerce recommendation [25]. Model compression approaches include knowledge distillation [26], lightweight network [27,28], low-rank approximation [29], quantization [30], and model pruning [31]. Knowledge distillation requires substantial resources and time to train a highly accurate large model to enhance the performance of small models. Lightweight networks have a streamlined structure, but the simplification of the structure can lead to performance decline. Low-rank approximation and quantization methods compress models by decomposing the model weight matrix and reducing the numerical precision of the model parameters, respectively. These two methods may impact model performance, but the effects can be recovered through fine-tuning. Compared to low-rank approximation and quantization, model pruning offers an additional advantage of high interpretability.

Therefore, our paper will focus on exploring model pruning. Model pruning is classified into structured [32,33] and unstructured types [34,35]. Unstructured pruning removes unimportant neurons and connections, thereby altering the model’s structure. This structural change cannot enable the model to directly accelerate on existing hardware. Structured pruning, achieved by removing entire operators from the model, preserves the model’s structural integrity and allows for hardware acceleration through devices such as GPUs. To fully utilize the parallel computing capabilities of hardware, structured pruning becomes our choice. The application of structured pruning technology in seed detection research is becoming increasingly widespread. Li et al. [36] propose a new two-dimensional information entropy filter pruning algorithm to achieve structured pruning. The pruned model achieves a single image inference speed of 107 FPS (frames per second), with the best accuracy of 95.94% on the red kidney bean dataset. Jin et al. [37] use the YOLOv5 target detection algorithm to obtain sunflower seeds from recorded videos and compare the effects of different degrees and methods of pruning on model performance. Wang et al. [38] train the improved YOLOv3 [39] model on a dataset containing 10 kidney bean seed varieties. Subsequently, the model is pruned using the scaling factors from the Batch Normalization layers [40] as a metric for channel importance. After fine-tuning the model with knowledge distillation, the number of model parameters is reduced by 98%, and the detection time is shortened by 59%.

Structured pruning methods can be divided into data-dependent [41,42,43,44,45] and data-independent [46,47,48,49] types. Although data-independent pruning is easy to implement, data-dependent pruning leverages actual training data to identify critical structures, thus maintaining higher accuracy and performance in the pruned model. Additionally, data-dependent pruning methods can prune according to specific data distributions and task requirements, making the pruned model more robust when handling similar data. Therefore, in the context of identifying rapeseed varieties, data-dependent pruning is superior. To this end, we propose a data-dependent two-dimension pruning method aimed at ensuring both the accuracy and real-time performance of the model. By sequentially pruning the spatial and channel dimensions of the baseline model, the model not only effectively reduces the number of parameters but also retains excellent performance. Ultimately, the optimized model achieves a mAP of 96.89%, and its inference speed on the Raspberry Pi is 1.18 s per image, meeting the real-time detection requirements for embedded devices.

The rest of the paper is organized as follows. Section 2 provides details of the created dataset. Section 3 describes our model design strategy. Section 4 evaluates our designs with experiments. Section 5 and Section 6 give the discussion and conclusion of this study, respectively.

## 2. Dataset Creation

In this study, the datasets involves five varieties of rapeseeds: Yuyou-35, Yuhuang-20, Deyou-5, Yuyou-55 and Yuyou-56. Example images of the rapeseeds are shown in Figure 1, and detailed information about the seeds is presented in Table 1. The rapeseeds of these five varieties are similar in terms of color, size, surface smoothness, and sphericity. It is obvious that discerning the varieties of these rapeseeds through human visual inspection is a considerable challenge for individuals without expertise, especially when one or a few seeds are singled out for identification.

To create the image dataset, we design a simple system of capturing rapeseed image, which mainly includes an Apple 14 Pro Max phone and a ring LED light (adjustable between 6.5 W and 15 W). For each image taken, we randomly select 4 to 40 seeds and place them on a sheet of white paper that is located 7 cm beneath the LED lights. Some of the seeds may be in contact with each other, but we do not deliberately separate them when capturing images. Our arrangement of seeds is consistent with the way seeds are positioned in real-world applications for identifying seeds on a conveyor belt. In total, 1871 images with a pixel size of 1280×1707 are taken and saved in JPG format. The acquired dataset is randomly divided into a training set (80%), a validation set (10%), and a test set (10%). The images used for training and validation do not overlap to ensure the reliability of subsequent evaluations.

During the model’s data loading process, data augmentation strategies are applied to increase image diversity. Specific augmentation strategies include random adjustments of hue (ranging from −0.015 to 0.015), saturation (ranging from 0.7 to 1.7) and brightness (ranging from 0.6 to 1.4); random scaling of images (with scales ranging from 0.8 to 1.2); and the use of the Mosaic method to combine four images into one. These data augmentation methods help the model adapt to different lighting conditions and object distances, thereby improving the generalization ability and robustness of the model. The dataset size stays the same after data augmentation, but the internal diversity of each image is significantly increased, which enables the model to encounter more diverse scenarios during the training process.

## 3. Model Design

### 3.1. Overview

We develop a rapeseed varieties detection system, the heart of which is an object detection DNN model. Its workflow is illustrated in Figure 2. The system uses a ring LED light combined with a camera lens to capture images of seeds. Then, the image is transmitted in real-time to an embedded device for processing. The embedded device runs the object detection model and then presents the detection results on an integrated display screen. In the detection results, seeds are highlighted with bounding boxes, and their varieties are labeled in the top-right corner of each box.

Our seed detection model is based on YOLOv7. Although YOLOv7 can achieve desirable accuracy on embedded devices, its slow inference limits its practicality in real-world seed detection. To this end, we apply model pruning techniques to the original YOLOv7 model, resulting in a streamlining YOLOv7 that significantly reduces the model size and computational complexity while maintaining high detection accuracy. With doing so, we develop a lightweight and accurate detection model that integrates seamlessly into our system, enabling the rapid identification of rapeseed varieties.

### 3.2. Pruning Strategies

#### 3.2.1. Pruning Process

The pruning process of the model training phase is shown in Figure 3. Following over-parameterization, YOLOv7 initially undergoes a conventional training for 400 epochs to achieve a high accuracy. Subsequently, the model engages in sparse training, which refines the model architecture by adjusting the distribution of weights. After sparse training, we obtain a set of optimized weights. Based on optimal weights, the model enters the pruning phase. Spatial pruning prunes the unimportant branches within the over-parameterized modules of the model. Channel pruning prunes the unimportant channels of the convolutional layers. Although the pruning process is divided into two parts, they are completed at once. Model pruning changes the structure of the model, which will lead to accuracy decline. In order to adapt the model to this change, we use the fine-tuning operation to recover model accuracy.

#### 3.2.2. Spatial Pruning

To enhance our model’s generalization ability and to better capture the complexity and diversity of the data, we over-parameterize the MP module into a multi-branch module (marked as MB module) prior to model pruning. The over-parameterization process is shown in Figure 4. The MB module enables different branches to learn distinct feature representations and integrates them together. Figure 4b illustrates the specific process of over-parameterization. The blue square represents the input tensor of a certain layer of the model, and four parallel operational branches extend from it. The architecture of these four branches is as follows: the first one is convolution followed by max-pooling, the second one is max-pooling followed by convolution, the third one consists solely of convolution, and the fourth one includes convolution followed by upsampling. Each branch focuses on different types of features or different spatial scales. Such a modification helps improve the model performance in dealing with fine-grained object detection tasks. Although each branch employs a different strategy when processing the input data, their final outputs maintain a consistent size. This design allows the model to easily merge these outputs, thus utilizing information from various scales and features. In the MP module, concatenation is used to merge the two channels, while preserving all information from both channels and allowing the model to learn comprehensive data features. In the MB module, we use addition to merge branches, as it avoids generating extra channels and does not require channel reduction during pruning. Therefore, the use of addition not only simplifies the pruning process but also improves the efficiency of our model.

During the sparse training process of an over-parameterized model, backpropagation adjusts the parameter δ to minimize the loss function. We use the value of δ to measure the contribution of each branch to the results (if a branch contributes more, δ will be larger; if its contribution is less, δ will be smaller). After training, we select the branch with the highest contribution (the most important) to replace the MB module in the over-parameterized model.

#### 3.2.3. Channel Pruning

In convolutional neural networks, the number of channels refers to the number of feature maps produced by convolutional layers. We prune the channel dimension using a Batch Normalization (BN) pruning strategy. The role of the BN layer is to normalize the output of each channel and then restore the scale and position of the data through scaling and shifting. If the λ value of a channel is large, it means that the features of this channel are amplified after normalization, indicating that the model considers the information in this channel to be very important. Conversely, a smaller λ value implies that the channel is less important. The BN pruning strategy accurately identifies and removes redundant channels by analyzing the scaling factor λ. Thereby, setting appropriate λ thresholds to guide pruning not only improves the performance and inference speed of the model but also serves as a regularization that prevents overfitting.

### 3.3. Model Structure and Training

#### 3.3.1. Model Structure

The overall architecture of our model is shown in Figure 5. The dashed box ① represents the process of spatial pruning: after over-parameterizing the MP module into the MB module, three branches are removed, and only one key branch is retained to replace the original MP module in the architecture. Dashed Box ② represents the process of channel pruning, where gray parts represent the removed channels while green parts represent the retained channels. After channel pruning, unimportant channels and their connections are removed from the model.

#### 3.3.2. Sparse Training

After training the over-parameterized model using transfer learning for 400 epochs, we apply sparsity inducing regularization to the model. The over-parameterized model learns redundant features, which interfere with the learning of genuine data features and consequently impact model performance and accuracy. However, sparsification alters the parameter distribution of the model, causing it to focus more on learning critical features. To achieve optimal model performance, we examine several critical hyperparameters based on experience and experiments. The empirically optimal hyperparameters are listed in Table 2. Our model is developed in Python 3.8 and trained on a computer equipped with an RTX 4090 GPU (24 GB) and an Intel® Xeon® Gold 6430 CPU.

#### 3.3.3. Loss Function

We design an objective function to perform sparse training of an over-parameterized model. The first term of Equation (Equation 1) represents the normal loss function of YOLOv7. $L_{Sp}$ and $L_{Ch1}$ represent the loss in the spatial and channel dimensions of the MB module, respectively. $L_{Ch2}$ represents the channel dimension loss of other convolutional layers. *a*, *b* are used to adjust the weight coefficients of the three in the total loss, and their values are shown in Table 2:(1)$$Loss=L+aL_{Sp}+b\left(L_{Ch1}+L_{Ch2}\right)$$

$L_{Sp}$ (Equation (Equation 2)) represents the regularization term for branch importance scores in the MB module. $N_{1}$ represents the number of MB modules in YOLOv7, and $N_{B}$ represents the number of branches in every MB module, which is 4. The output of each branch in the MB module is multiplied by an importance score. This importance score is computed using the function S(·), which is the sigmoid function (Equation (Equation 3)). To better adapt the sigmoid function to the variations in importance scores, a learnable parameter δ is used. This parameter is learned during the sparse training process through backpropagation. The value of δ adjusts according to the gradient of the loss function, making the output of the sigmoid function better correspond to the importance of each channel. By incorporating δ into the regularization term of the loss function, we balance the importance of each branch, thereby obtaining a more stable model:(2)$$L_{Sp}=\sum_{l=1}^{N_{1}}\sum_{i=1}^{N_{B}}S\left(\delta_{i}^{l}\right)$$
(3)$$S\left(\delta_{i}\right)=\frac{\exp\left(\delta_{i}\right)}{\sum_{b=1}^{N_{B}}\exp\left(\delta_{b}\right)}$$

In Equation (Equation 4), $L_{Ch1}$ is the L1 regularization constraint on the λ coefficient in the BN layer in MB module. Here, $N_{C}$ represents the number of convolutional layers in each branch. $z_{out(i,j)}^{\left(l\right)}$ represents the number of output channels in the *j*-th layer of the *i*-th branch in the *l*-th MB module. ∥·∥ represents the L1 norm. Each channel has a scale factor λ, which is used to adjust the weight of the channel after pruning. λ is a parameter in the Batch Normalization process (as shown in Equation (Equation 5)). In Equation (Equation 5), B represents the current batch, $\mu_{B}$ and $\sigma_{B}$ represent the mean and standard deviation of the batch, respectively, and ϵ is a very small constant. λ and β are trainable scale and shift parameters in the affine transformation. $\lambda_{\left(i,j,k\right)}^{\left(l\right)}$ represents the λ value corresponding to the *k*-th channel in the *j*-th layer of the *i*-th branch in the *l*-th MB module. After the model undergoes sparse training, the larger the λ value, the more important the channel:(4)$$L_{Ch1}=\sum_{l=1}^{N_{1}}\sum_{i=1}^{N_{B}}\sum_{j=1}^{N_{C}}\sum_{k=1}^{{Z_{out}}_{\left(i,j\right)}^{\left(l\right)}}∥\lambda_{\left(i,j,k\right)}^{\left(l\right)}∥$$
(5)$$\hat{Z}=\frac{{Z_{in}}_{\left(i,j\right)}^{\left(l\right)}-\mu_{B}}{\sqrt{\sigma_{B}^{2}+\epsilon }},{Z_{out}}_{\left(i,j\right)}^{\left(l\right)}=\lambda \hat{z}+\beta$$

Equations (Equation 6) and (Equation 7) represent the sparsity regularization formulas for convolutional layers other than the RepConv, CBM and MB modules. The number of these convolutional layers is denoted as $N_{2}$. The pruning principle is also BN pruning, so the loss item logic is similar to Equations (Equation 4) and (Equation 5). Slightly differently, ${Z_{out}}^{\left(l\right)}$ represents the number of output channels in the *l*-th convolutional layer, and $\lambda_{k}^{\left(l\right)}$ represents the scale factor corresponding to the *k*-th channel in the *l*-th layer:(6)$$L_{Ch2}=\sum_{l=1}^{N_{2}}\sum_{k=1}^{{Z_{out}}^{\left(l\right)}}∥\lambda_{k}^{\left(l\right)}∥$$
(7)$$\hat{Z}=\frac{{Z_{in}}^{\left(l\right)}-\mu_{B}}{\sqrt{\sigma_{B}^{2}+\epsilon }},{Z_{out}}^{\left(l\right)}=\lambda \hat{z}+\beta$$

### 3.4. Prototype Implementation

We implement a prototype of our system which is shown in Figure 6. It centers a Raspberry Pi 4B device, a typical low-cost embedded computing device. Raspberry Pi encapsulates an ARM micro-controller and offers a diverse range of hardware interfaces, enabling easy connections with external sensors or actuators, such as temperature sensors, cameras, motors, and so forth. Moreover, Raspberry Pi can run common embedded operating systems (e.g., embedded Linux and RTOS), which facilitate the development and expansion of user programs. The Raspberry Pi 4B of our prototype is equipped with a 1.5 GHz 64-bit quad-core processor and 8 GB RAM.

As depicted in Figure 6, the camera is placed in the center of the ring LED light to capture the seed to detect. With a resolution of 1080p (1920 × 1080 pixels), the camera is connected, via a USB cable, to the Raspberry Pi, which powers the camera and retrieves images to local memory.

After receiving images captured in YUV format, the Raspberry Pi first converts them to JPG format, and then processes them using the streamlining YOLOv7 model to obtain results. The process of the detection includes image capture, image storage, image preprocessing, object detection, result retrieval, and result presentation, and it repeats until the user chooses to stop.

To enhance the inference efficiency, we convert the model weight files (.pt format) to the ONNX (Open Neural Network Exchange) format for deployment. The ONNX format allows the model to utilize the ONNX Runtime inference engine for computation, which provides good compatibility with a variety of hardware accelerators. The change in format greatly improves the model performance on the Raspberry Pi.

## 4. Results

### 4.1. Metrics

We use performance metrics, including mAP (mean Average Precision), parameter count, and inference time on the Raspberry Pi 4B, to evaluate our system performance. The metric of mAP is calculated by
(8)$$\begin{array}{ccc} \mathrm{Precision} & = & \frac{\mathrm{TP}}{\mathrm{TP}+\mathrm{FP}} \end{array}$$
(9)$$\begin{array}{ccc} \mathrm{Recall} & = & \frac{\mathrm{TP}}{\mathrm{TP}+\mathrm{FN}} \end{array}$$
(10)$$\begin{array}{ccc} \mathrm{AP} & = & \int_{0}^{1}\mathrm{Precision}\left(\mathrm{Recall}\right)d\left(\mathrm{Recall}\right) \end{array}$$
(11)$$\begin{array}{ccc} \mathrm{mAP} & = & \frac{1}{c}\sum_{i=1}^{c}{\mathrm{AP}}_{i} \end{array}$$
where TP, FP, and FN represent true positives, false positives, and false negatives, respectively. Precision (Equation (Equation 8)) measures the proportion of true positive results among all positive results, while recall (Equation (9)) measures the proportion of true positive results that have been correctly identified. AP (Equation (10)) is the average precision for each category, calculated from the area under the precision–recall curve. The AP score provides an overall measure of model quality. mAP (Equation (11)) is the mean AP value for each category (c), used to measure the accuracy of an object detection algorithm across all categories. In our paper, when we mention mAP, specifically refer to mAP_0.5. This indicates that the evaluation metric we use is the mAP at an Intersection over Union (IoU) threshold of 0.5.

### 4.2. Comparison of Different Target Detection Algorithms

To select the baseline model, we compare the performance of YOLOv7x, YOLOv7-tiny, and YOLOv7. The results are shown in Table 3. YOLOv7x achieves a mAP of 95.70% on the dataset, with the slowest inference speed of 7.56 s among the three models, and a larger number of parameters at 70.84 M. This suggests that YOLOv7x is relatively complex for the rapeseed dataset used in this experiment, leading to overfitting. YOLOv7-tiny, a lightweight variant of YOLOv7, slightly outperforms YOLOv7x in accuracy at 96.15%, with an outstanding inference speed of 0.60 s and the lowest number of parameters at 6.02 M. Subsequent model pruning operations can also reduce model size and response time, so mAP becomes the primary consideration when selecting a baseline model. In this regard, YOLOv7 performs exceptionally well, achieving the highest mAP of 96.68% despite having an inference speed of 4.48 s and a parameter size of 36.50 M, both higher than those of YOLOv7-tiny. As the model with the highest mAP, YOLOv7 is chosen as the baseline model. The performance of the over-parameterized model YOLOv7 (MB) is shown in the last row of the table. Compared to the baseline model, the number of parameters is increased by 0.55 M, while the mAP is only decreased by 0.02%, and the inference speed is only increased by 0.04 s.

### 4.3. Comparison of Different Branch Selections

To verify the accuracy and effectiveness of the selected branches in the MB module, we design a set of comparative experiments based on YOLOv7 (MB). We only reduce the parameters in the spatial dimension through branch selection, without making any changes to the parameters in the channel dimension. Table 4 presents the performance comparison between the spatial dimension branch selection method and the random branch selection method. The results indicate that the mAPs of the two models with randomly selected branches converge to over 96.6%. However, this result still falls short of the mAP achieved by the spatial dimension branch selection method, which is 96.81%. This demonstrates that the branch selection method for the MB module is accurate and effective during the spatial dimension pruning process.

### 4.4. Comparison of Different Channel Dimension Pruning Methods

#### 4.4.1. Global Pruning

After sparse training, we plot the distribution graph of the scaling factors of all channels in the layers available for pruning (Figure 7). To enhance the visualization effect, all layers are divided into two groups. The channels in the first few layers have higher scaling factor values. The median of the scaling factors of the first four layers is all greater than 0.7. It is because the initial layers of the model directly process raw data and extract information such as edges, colors, and textures. These layers provide the necessary foundational information for further processing, which is crucial for the overall performance of the entire model. To avoid pruning all parameters of a layer, we rank the highest scaling factor of each layer, and use the smallest value among them as the threshold to define the ratio of global pruning. In this experiment, the threshold is 0.622, which means that up to 62.2% of the parameters can be pruned. We adjust the pruning ratio to explore the performance of global pruning.

As shown in Table 5, the inference speed and the number of parameters decrease as the pruning ratio increases. The mAP fluctuates slightly after pruning at various ratios compared to before pruning, and remains at a high level. This suggests that the feature expression ability of each layer of the model is uneven, and global pruning cannot achieve the best effect. Therefore, we opt for the layer-by-layer pruning strategy.

#### 4.4.2. Fixed Ratio Layer-by-Layer Pruning

Fixed ratio layer-by-layer pruning means pruning a fixed ratio of parameters for each layer. Figure 8 shows the change in mAP with the number of training iterations for YOLOv7 (MB) under different pruning ratios (0.3–0.7). It can be observed that before the 150-th epoch, the convergence speed of the curves increases with pruning ratios of 0.3, 0.4, and 0.5. This is because the model is gradually pruned of redundant connections that have little impact on model performance, which does not significantly affect the accuracy of the model. And, due to the reduction in model complexity, the convergence speed actually increases. After 150 epochs, the mAP tends to stabilize and converges around 96.8%. But the model experiences oscillations when pruned at 0.6 and 0.7 and ultimately does not achieve convergence. The results suggest that pruning causes the model to lose some important information, making it difficult for the model to achieve a stable state during the training process. As can be seen from Figure 7, the overall importance factor of some layers is significantly higher than other channels. Pruning these channels at a high ratio leads to a decline in model performance. So, fixed ratio layer-by-layer pruning is not applicable in this experiment.

#### 4.4.3. Custom Ratio Layer-by-Layer Pruning

Global pruning and fixed ratio layer-by-layer pruning are relatively limited and may ignore the features of the model. So, we consider the custom ratio layer-by-layer pruning, which means applying different pruning ratios to each layer. This allows us to flexibly adjust the pruning ratio between different layers, thus achieving finer control of the model. We specify smaller pruning ratios for layers with more important channels, and larger ratios for layers with fewer important channels. The important channels are those with a high proportion factor, and the distribution of channel scaling factors within each layer can be seen in Figure 7. Figure 9 shows the ratio of channels retained in each layer of the model, corresponding to the performance indicators in Table 6. As can be seen from the table, the mAP of ‘Custom1’ reaches 96.89%, which surpasses the baseline model. The inference speed decreases from 4.48 s to 1.18 s, and the number of parameters drops from 36.50 M to 9.19 M, compared to the baseline model. ‘Custom2’ and ‘Custom3’ respectively reduce the ratio of retained channels in layers with fewer and more important channels. The two have similar inference speeds, but the mAP differs by 0.65%. This suggests that the method of judging channel importance in our paper is effective (models that prune less important channels are more accurate than models that prune more important channels). ‘Random’ randomly selects the pruning ratio for each layer. After the model converges, the mAP only reaches 95.95%, which is lower than the others, and the inference speed is also slower. Considering the above four pruning schemes, we believe that the ‘custom1’ scheme has the highest mAP and a reasonably fast inference speed, and offers the best performance.

## 5. Discussion

The real-time detection of rapeseeds on an embedded computing environment is highly valuable in the agricultural industry. Our study introduces the dual-dimensional pruning to YOLOv7, aimed at accelerating its inference speed on embedded devices without compromising accuracy.

Our paper draws inspiration from [50], which aims to over-parameterize every convolutional layer of the model. Our method combines the structural characteristics of YOLOv7, merely over-parameterizing the MP module. There are two reasons for the modification. First, the scaling factors of most layers in our model tend to be zero, resulting in a large number of the additional parameters being pruned later on. Therefore, it is unnecessary to over-parameterize every convolutional layer. Second, the rapeseed dataset used in this experiment is moderate in size. If the model is over-parameterized into a super-large model, it can easily lead to overfitting, and the weights of the model may not be optimal after training convergence. To validate the performance of our model, we construct a seed detection system. The seed detection system only requires a low-cost commercial camera, an LED light, and an embedded computing device to achieve quick and high-accuracy variety detection. The adaptability of our model ensures that the system can be flexibly deployed across a range of scenarios, from laboratories and fields to remote outdoor settings.

However, our model is currently limited in the range of seed varieties it can detect, which restricts its application in actual agricultural production and seed screening tasks. To further improve the adaptability of the system, it is necessary to use a diverse seed dataset to train the model. By enriching the seed dataset, the detection range of the model can be expanded, and the robustness and accuracy of the model can be improved.

## 6. Conclusions

This study aims to develop an effective and efficient detection system for rapeseed varieties that can be deployed on embedded devices with limited computation resource. We collect five varieties of rapeseeds as samples and use an image acquisition device to obtain 1871 images. To improve real-time performance, we use pruning techniques to lighten the model. Specifically, a dual-dimension pruning method is designed, which can reduce redundant parameters in both spatial and channel dimensions. We demonstrate the effectiveness of spatial pruning by comparing it with models that randomly select branches. We also compare three channel pruning methods, among which custom ratio layer-by-layer pruning is more precise and flexible, achieving the best model performance. Finally, our model achieves a 96.89% mAP with only 9.19 M parameters, and an inference speed of 1.18 s per image, meeting the requirements for real-time detection of rapeseeds in resource-constrained scenarios. In the future, we will plan to introduce layer dimension based on the existing dual-dimension pruning, to achieve finer-grained model adjustments. This will provide greater potential and flexibility for the application of target detection models on embedded devices.

## Figures and Tables

**Figure 1 sensors-24-05585-f001:**
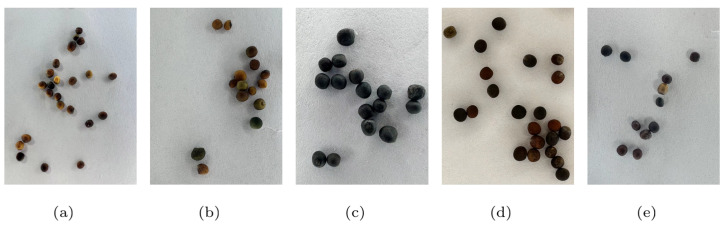
Five varieties of rapeseed samples. (**a**) Yuyou-35; (**b**) Yuhuang-20; (**c**) Deyou-5; (**d**) Yuyou-55; (**e**) Yuyou-56.

**Figure 2 sensors-24-05585-f002:**
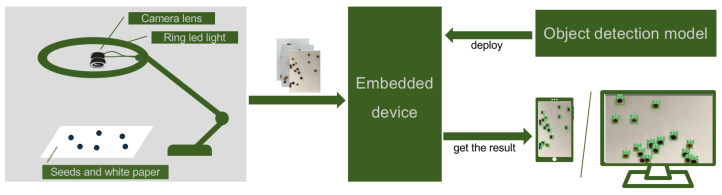
Workflow of our system.

**Figure 3 sensors-24-05585-f003:**
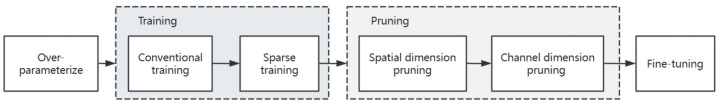
The pruning process of the model training phase.

**Figure 4 sensors-24-05585-f004:**
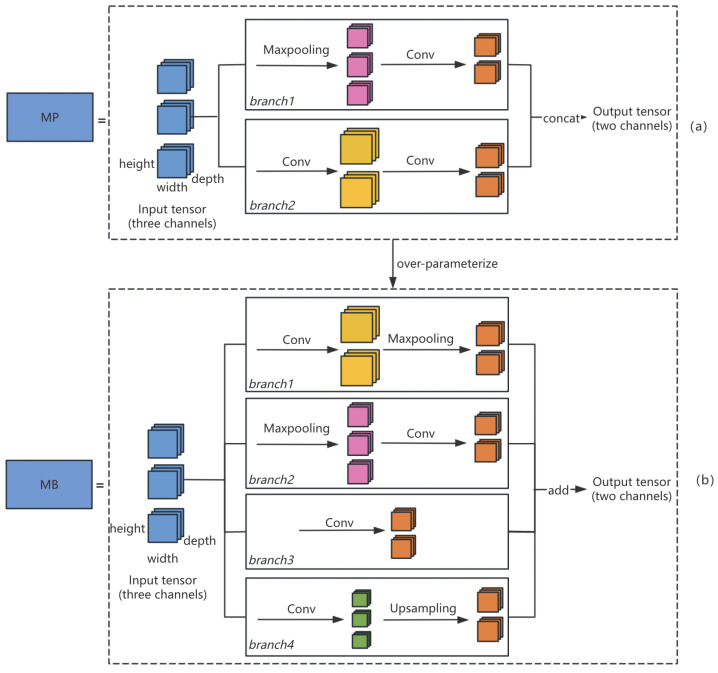
Over-parameterization in spatial pruning from the MP structure (**a**) to the MB structure (**b**).

**Figure 5 sensors-24-05585-f005:**
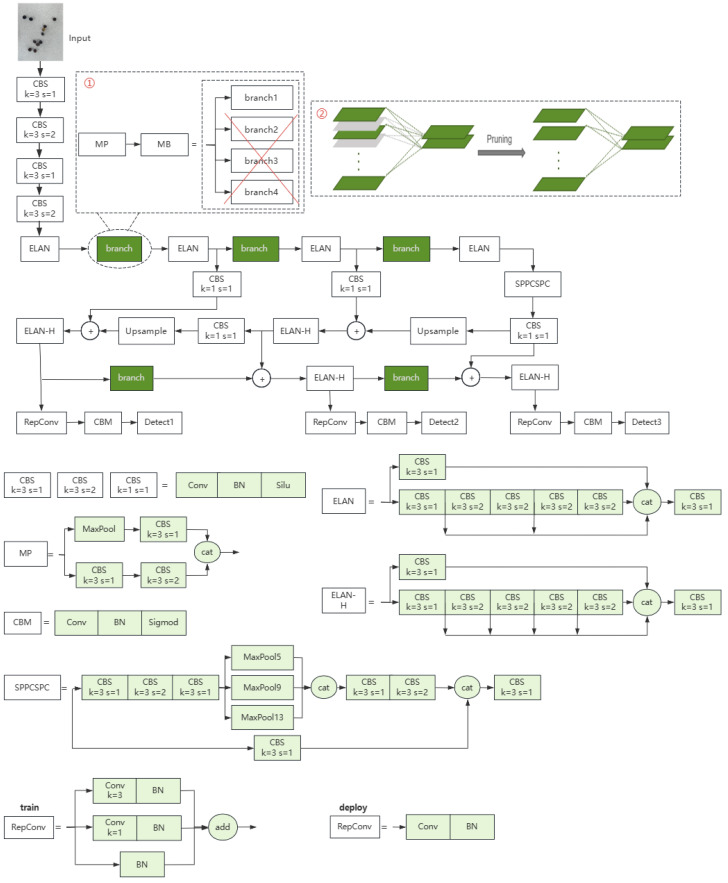
Architecture of our model. The specific structures of the modules used in the architecture are listed in the figure. The ‘branch’ module is the best branch among the four branches in MB, and its specific structure is shown in Figure 4.

**Figure 6 sensors-24-05585-f006:**
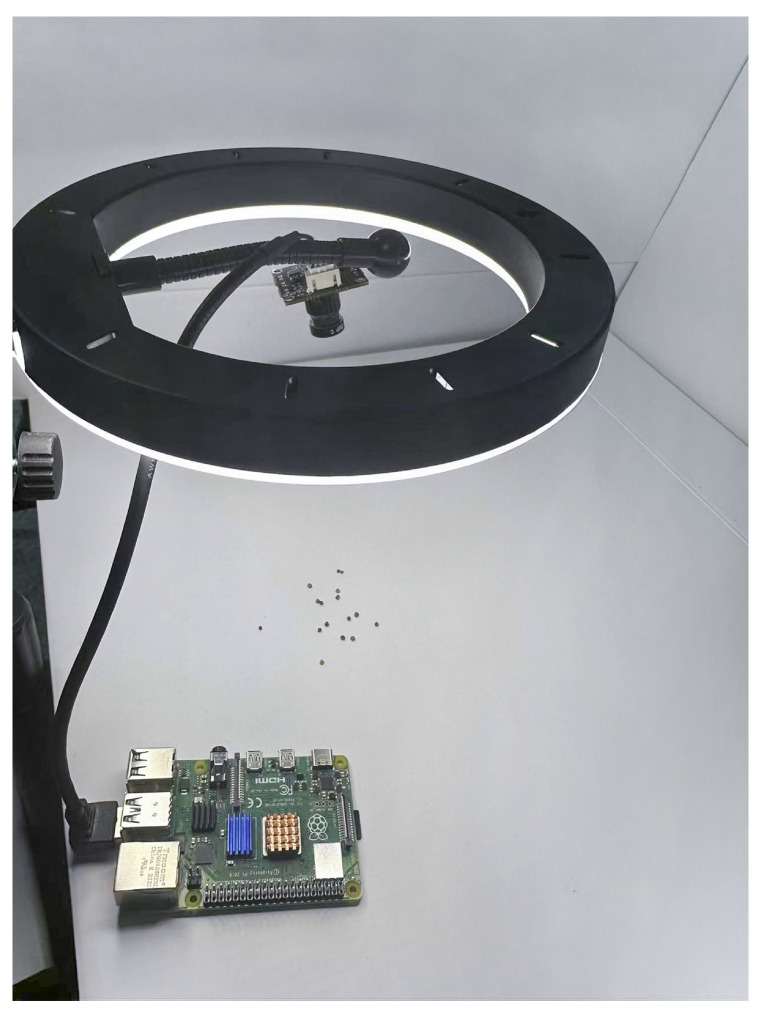
Seed variety detection device.

**Figure 7 sensors-24-05585-f007:**
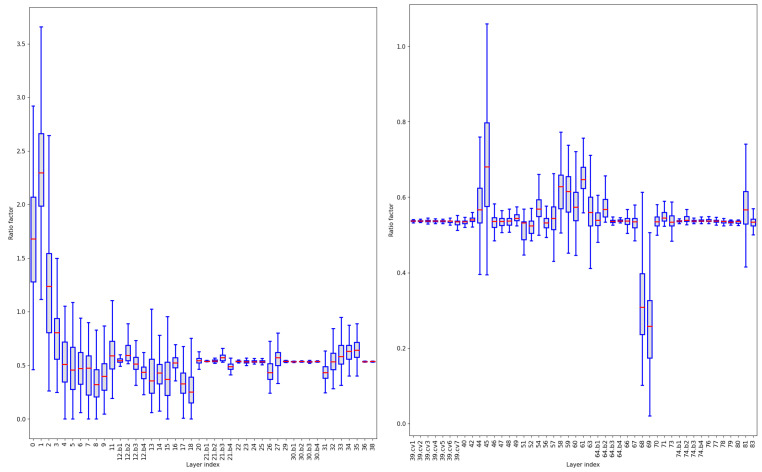
Scaling factor distributions in YOLOv7 (MB). For display convenience, it is divided into left and right graphs.

**Figure 8 sensors-24-05585-f008:**
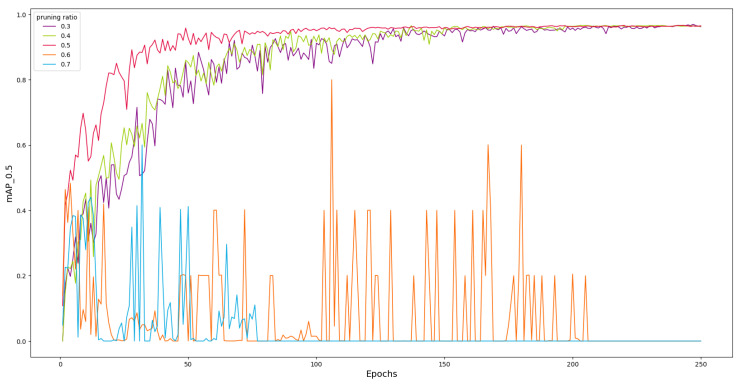
Influence of different pruning ratios on mAP.

**Figure 9 sensors-24-05585-f009:**
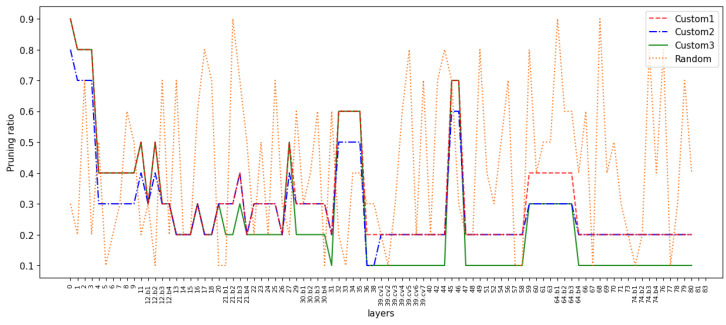
Channel retention ratio of each layer. The solid lines represent three custom pruning schemes, and the dashed line represents the random pruning scheme.

**Table 1 sensors-24-05585-t001:** Information of five varieties of rapeseeds.

Variety	Registration Year	Origin	Color	Quantity
Yuyou-35	2019	GH31 × GH33	tawny	345
Yuhuang-20	2022	W01YA × L804	tawny	351
Deyou-5	2019	K97A × Z68R	dark brown	356
Yuyou-55	2021	17,346 × 16,110	dark brown	414
Yuyou-56	2022	17448A × 18,007	ark brown	405

**Table 2 sensors-24-05585-t002:** Hyperparameter settings for sparse training.

Hyperparameters		Value
weight coefficients	a	0.0001
	b	0.0001
initial learning rate		0.01
batch size		16
#epochs		400

**Table 3 sensors-24-05585-t003:** Comparison of model performance. The first three rows show the performance metrics of YOLOv7x, YOLOv7-tiny, and YOLOv7, while the last row shows the performance metrics of YOLOv7 (MB).

Models	mAP (%)	Inference Speed (s)	Number of Parameters (M)
YOLOv7x	95.70	7.56	70.84
YOLOv7-tiny	96.15	0.60	6.02
YOLOv7	96.68	4.48	36.50
YOLOv7 (MB)	96.66	4.52	37.05

**Table 4 sensors-24-05585-t004:** Performance comparison of our method and random branch selection method. It shows the branch selections for five MB modules, which all possess the same structure. bn (n ∈ {1, 2, 3, 4}) represents the n-th branch.

Branch Selection Methods	Branch Selections					mAP (%)
	MB1	MB2	MB3	MB4	MB5	
pruning strategy selection	b2	b3	b1	b3	b4	96.81
random selection1	b3	b4	b2	b1	b3	96.63
random selection2	b1	b2	b3	b4	b2	96.76

**Table 5 sensors-24-05585-t005:** Comparison of performance of different ratios of pruning.

Pruning Ratios	mAP (%)	Inference Time (s)	Number of Parameters (M)
0.1	96.81	3.35	29.20
0.2	96.79	2.94	24.25
0.3	96.75	2.73	19.24
0.4	96.80	2.54	18.16
0.5	96.73	2.34	15.34
0.6	96.64	2.25	13.24
0.622	96.53	2.14	13.06

**Table 6 sensors-24-05585-t006:** Comparison of model performance. The first three rows represent three custom pruning schemes, and the last row represents a random pruning scheme.

	mAP (%)	Inference Speed (s)	Number of Parameters (M)
Custom1	96.89	1.18	9.19
Custom2	96.12	1.17	8.92
Custom3	96.77	1.15	8.77
Random	95.94	1.33	13.15

## Data Availability

The raw data supporting the conclusions of this article will be made available by the authors on request.

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
