# Peer review of "Streamlining YOLOv7 for Rapid and Accurate Detection of Rapeseed Varieties on Embedded Device"

_sensors, 2024, doi:10.3390/s24175585_

Round 1

Reviewer 1 Report

Comments and Suggestions for Authors

1. The main improvement described in the paper is spatial dimension pruning, but this is not reflected in the title of the article.

2. In the Introduction, the pruning technology can be discussed with a larger proportion, and more citations related to pruning technology can be selected. About YOLOv7x、YOLOv7-tiny and YOLOv7 should be introduced in introduction.

3. In the’ Dataset Creation how many instances are there in each image? Have you considered creating datasets in complex environments, or denser datasets?

4. In Model Design, the description of Yolov7 can be put into introduction or deleted directly. 

5. The layout of tables still needs improvement; The best results or the results that require more attention should be labeled.

6. In fig.3 the title is not enough to express the graph.

7. In fig.4, From MP to MB, It implies a flow ,maybe an arrow should be added.

8. In fig.5 It is very difficult to understand such as ELAN or CBS,CBM. And Branch?

9. In fig.6,(b) is too easily express. An new style can be selected.how to understand streamlining YOLOV7?

Comments on the Quality of English Language

1. Please unify the words in the article. For example,Figure 4 is used in the caption while Fig. 4 is used in the text; Table 4 is used in the note in the caption while Tab.4 is used in the text.

2. There are some grammatical errors in the article, please fix them.

Author Response

Thank you very much for taking the time to review this manuscript.

Comments 1: The main improvement described in the paper is spatial dimension pruning, but this is not reflected in the title of the article.

Response 1: Thank you for pointing this out. We understand the importance of accurately reflecting the main contributions of the paper in the title. However, we believe that the current title effectively captures the essence of our work. Our research focuses on using pruning techniques to streamline the YOLOv7 model, making it more efficient for deployment on embedded devices. We choose the term “streamlining” in the title to broadly convey this optimization, which can attract a wider audience, including those who may not be familiar with pruning techniques. They can gain a deeper understanding of the specific methods through the paper.

Comments 2:In the Introduction, the pruning technology can be discussed with a larger proportion, and more citations related to pruning technology can be selected. About YOLOv7x、YOLOv7-tiny and YOLOv7 should be introduced in introduction.

Response 2: Thank you for pointing this out, we agree with this comment. Therefore,in the second sentence of the sixth paragraph of the Introduction, we add references to papers on structured and unstructured pruning. In the seventh paragraph of the Introduction, we introduce the applicability of data-dependent dual-dimensional pruning methods to application scenarios. For the descriptions related to YOLOv7x, YOLOv7-tiny and YOLOv7, we add the fourth paragraph in the Introduction.

Comments 3: In the’ Dataset Creation how many instances are there in each image? Have you considered creating datasets in complex environments, or denser datasets?

Response 3:

Thank you for your comment. 

First, the second sentence of the second paragraph within ‘Dataset Creation’ section details that, during the image capture process, between 4 and 40 rapeseed grains were positioned on white paper, indicating that each image comprises 4 to 40 instances.

Second, during the preparation phase of data collection, we considered the possibility of capturing images in complex environments. However, we ultimately chose to shoot against a single background for the following reasons: 1. The high mobility of rapeseeds allows users to easily place them on a simple background for shooting, thus reducing environmental interference. 2. In an automated industrial environment, rapeseeds are usually processed and sorted on a conveyor belt, and using a single background can effectively simulate this situation. If future application scenarios require, we can consider expanding the dataset to include image capture under more complex environments. Additionally, our dataset includes instances where rapeseeds are in close contact, which simulates dense seed scenes. This setup provides practical experience for the model training to handle dense seed layouts.

Comments 4: In Model Design, the description of Yolov7 can be put into introduction or deleted directly. 

Response 4: Thank you for pointing this out, we agree with this comment. Therefore,we delete the detailed description of YOLOv7 in Chapter 3 (Model Design) and move it to the fourth paragraph in the Introduction.

Comments 5: The layout of tables still needs improvement; The best results or the results that require more attention should be labeled.

Response 5: Thank you for pointing this out, we agree with this comment. We bold the rows that need to be highlighted in Tables 4 and 6, while the data in the other tables does not need to be bolded.

Comments 6: In fig.3 the title is not enough to express the graph.

Response 6: Thank you for pointing this out, we agree with this comment. To better convey the content, we change the title to “The pruning process of the model training phase.” This title more specifically describes the content of Figure 3 , allowing readers to more accurately understand the stages and focus presented in the figure.

Comments 7: In fig.4, From MP to MB, It implies a flow ,maybe an arrow should be added.

Response 7: Thank you for pointing this out, we agree with this comment. We update Figure 4 to more clearly illustrate the flow from MP to MB.

Comments 8: In fig.5 It is very difficult to understand such as ELAN or CBS,CBM. And Branch?

Response 8: Thank you for pointing this out, we agree with this comment. We expand Figure 5 to include detailed structures of the modules appearing in the architecture of the model (e.g., CBS, CBM, ELEN, etc.), and  add explanations of the ’branch‘ in the title.

Comments 9: In fig.6,(b) is too easily express. An new style can be selected.how to understand streamlining YOLOV7?

Response 9: Thank you for pointing this out, we agree with this comment. We delete Figure 6(b) and add the corresponding description to the third paragraph of the Prototype Implementation section in Chapter 3 (Model Design). Regarding how to understand streamlining YOLOv7, we revise the second paragraph of the Overview section in Chapter 3 (Model Design) to explain that streamlining YOLOv7 refers to the pruned model.

Comments 10:  Please unify the words in the article. For example,Figure 4 is used in the caption while Fig. 4 is used in the text; Table 4 is used in the note in the caption while Tab.4 is used in the text.

Response 10: Thank you for your suggestion. We substitute all “Fig.” in the text with “Figure.” and all “Tab.” with “Table.”

Reviewer 2 Report

Comments and Suggestions for Authors

1.The average precision mean (mAP) of the improved algorithm increased from 96.68% to 96.89%, but there was not much improvement;

2.The quality and diversity of the dataset are directly related to the effectiveness of model learning. The 1871 datasets mentioned in the article contain how many datasets for each of the five varieties? Have you considered performing data augmentation processing?

3.Is the workflow of the variety detection system designed in the article only for target detection? So how to distinguish different varieties?

4.What is the specific process of pruning? What is an unimportant branch? Suggest adding a theoretical analysis of feasibility.

5.Why are there two YOLOv7s in Table 3?

6.Compared to YOLOv7, YOLOv7 tiny has similar mAP and small model parameters. Why not directly improve YOLOv7 tiny to increase accuracy?

Comments on the Quality of English Language

Suggest finding a professional organization for English grammar proofreading

Author Response

Comments 1: The average precision mean (mAP) of the improved algorithm increased from 96.68% to 96.89%, but there was not much improvement;

Response 1: Thank you for pointing this out. Allow me to explain. Pruning is a technique used to prune redundant parameters to enhance the inference speed of the model. Therefore, it primarily optimizes speed rather than accuracy. Since pruning reduces overfitting, there is a passive improvement in accuracy. Although the mAP improvement is not significant, with our pruning method, the model’s inference speed improved significantly from 4.48 s to 1.18 s, which is a substantial enhancement.

Comments 2: The quality and diversity of the dataset are directly related to the effectiveness of model learning. The 1871 datasets mentioned in the article contain how many datasets for each of the five varieties? Have you considered performing data augmentation processing?

Response 2: Thank you for pointing this out. The quantities for the five varieties are listed in the last column of Table 1. Specifically, the dataset sizes for Yuyou-35, Yuhuang-20, Deyou-5, Yuyou-55, and Yuyou-56 are 345, 351, 356, 414 and 405, respectively. As for whether data augmentation is performed, the answer is yes. We add a third paragraph in the second chapter (Dataset Creation) to describe the data augmentation strategies used in this paper.

Comments 3: Is the workflow of the variety detection system designed in the article only for target detection? So how to distinguish different varieties?

Response 3: Thank you for pointing this out. The object detection model used in this paper detects not only the location of the objects but also their categories. We add a description of the model’s detection results at the end of the first paragraph of section 3.1(Overview).

Comments 4: What is the specific process of pruning? What is an unimportant branch? Suggest adding a theoretical analysis of feasibility.

Response 4: Thank you for pointing this out, we agree with this comment. In section 3.2 (Pruning Strategies) , we enrich the description of pruning details in the second paragraph of the ‘Spatial Pruning’ section and in the ‘Channel Pruning’ section.

Comments 5:Why are there two YOLOv7s in Table 3?

Response 5: Thank you for your question. I will explain this for you. The third row in the table 3 shows the performance metrics of the YOLOv7 model, and the fourth row shows the performance metrics of the YOLOv7 (MB) model. In the last two sentences of the section 4.2 (Comparison of different target detection algorithms), we introduce that YOLOv7 (MB) is the model after over-parameterizing YOLOv7. The process of over-parameterization is specifically described in the second chapter.

Comments 6:Compared to YOLOv7, YOLOv7 tiny has similar mAP and small model parameters. Why not directly improve YOLOv7 tiny to increase accuracy?

Response 6:

Thank you for pointing this out. Let me explain the rationale behind our choice. YOLOv7-tiny is designed to be highly compact, resulting in a limited number of parameters. Consequently, its potential as a baseline model is restricted. Any further reduction in parameters may significantly affect its accuracy and performance. In contrast, while the YOLOv7 model has more parameters, it offers higher training accuracy and stronger expressive capability. During training, YOLOv7 autonomously learns weight parameters better suited to the current dataset, and even after pruning, the model still maintains relatively high accuracy. Therefore, we do not choose to prune YOLOv7tiny.

Reviewer 3 Report

Comments and Suggestions for Authors

The authors proposed the adaptation of YOLOv7 which performs real-time detection of rapeseed varieties, to replace labor-intensive and time-consuming expert work. In YOLOv7, the authors replaced MP module with multi-branch module. Then, they explored several models of structured pruning of the network and selected the best one, to adapt it for a device with limited computational resources, lightening and accelerating the network without compromising its precision. The authors designed the loss function, with parameters used in the pruning strategy, which include the selection of the most efficient branch in the MB module, and channel pruning (custom ratio layer-by-layer pruning proved to be the most efficient). The significant strength of the paper is the proposal of the procedure that achieved: 1) to decrease 3.8 times the inference time (to the value of 1.18, which allows classification in real-time), 2) to decrease the number of parameters 4 times which is especially suitable for resource-constrained embedded devices while 3) the precision (mean average precision) increased by 0.23%.

To strengthen their work, it would be prudent, for the authors, to explain the different principles of channel pruning denoted as “Custom 1”, “Custom 2” and “Custom 3”. In other words, it is desirable that authors describe the way (principles, maybe mathematical expressions …) according to which the diagram in Figure 9 is created.

Similarly, for the sake of clarity, the differences between branch selection models MB1 to MB5 might be, in short, stated.

It might be considered whether it is worth sacrificing the stylish view of diagrams in figures 8 and 9, like they are now, to enhance their clarity by using lines of more intensive colors (apart from black, lines can be red, blue, green, orange, magenta …).

Suggestions for small changes:

In line 51, between references [14] and [15], the reference presenting the detection of various leguminous seeds is missing. I propose to add this reference in line 51, as well as in References section.

In Figures 4(a) and 4(b), instead of the word “weight”, the word “width” should be written.

In line 152, there is a typing error: in the word “modeule”, the first “e” should be deleted.

I suggest that authors reconsider the use of the words “sparse” and “sparsity”:

- in lines 170 and 202, “sparse training” might be more appropriate than “sparsity training”;

- in line 193, “sparsity inducing” might be more appropriate than “sparse inducing”.

In line 220, LBN1 in Eq. 4 is explained, whereas there is no LBN1 in Eq.4 in line 231. Maybe, replacing it with LCh1 is the appropriate correction.

Author Response

Comments 1: To strengthen their work, it would be prudent, for the authors, to explain the different principles of channel pruning denoted as “Custom 1”, “Custom 2” and “Custom 3”. In other words, it is desirable that authors describe the way (principles, maybe mathematical expressions …) according to which the diagram in Figure 9 is created.

Response 1: Thank you for pointing this out. In Section 4.4.3(Custom ratio layer-by-layer pruning), we add a fifth sentence for further clarification. After paragraph optimization, the first half of Section 4.4.3 details the design principles of the 'Custom 1', 'Custom 2' and 'Custom 3' pruning methods more explicitly. That is, the important channels are determined by scaling factors (the distribution of which is depicted in Figure 7), and the layer pruning ratios are decided based on the quantity of important channels in each layer.

Comments 2: Similarly, for the sake of clarity, the differences between branch selection models MB1 to MB5 might be, in short, stated.

Response 2: In the caption for Table 4, we add the description  which indicates the MB1-MB5 modules possess the same structure.

Comments 3: It might be considered whether it is worth sacrificing the stylish view of diagrams in figures 8 and 9, like they are now, to enhance their clarity by using lines of more intensive colors (apart from black, lines can be red, blue, green, orange, magenta …).

Response 3: Thank you for your suggestion. We agree with it. We change the line colors in Figures 8 and 9.

Comments 4: In line 51, between references [14] and [15], the reference presenting the detection of various leguminous seeds is missing. I propose to add this reference in line 51, as well as in References section.

Response 4: Thank you for your suggestion. In the third paragraph of the introduction, we expand the description of the experiments involving leguminous seeds, and the revised sentences appear in lines 50, 51, and 52 of the new manuscript. Please review.

Thank you for your language suggestions. It is very important to us. We revise the article based on the suggestions you provide:

Comments 5: In Figures 4(a) and 4(b), instead of the word “weight”, the word “width” should be written.

Response 5: We change “weight” to “width” in Figures 4(a) and 4(b).

Comments 6: In line 152, there is a typing error: in the word “modeule”, the first “e” should be deleted.

Response 6: We correct “modeule” to “module” . Due to revisions in the manuscript, the original line 152 is now located at line 182.

Comments 7: I suggest that authors reconsider the use of the words “sparse” and “sparsity”:

- in lines 170 and 202, “sparse training” might be more appropriate than “sparsity training”;

- in line 193, “sparsity inducing” might be more appropriate than “sparse inducing”.

Response 7: We reconsider the use of “sparse” and “sparsity” in our manuscript as you highlight. In the term ‘sparsity inducing regularization,’ “sparsity” serves as an adjective that modifies ‘inducing regularization,’ emphasizing the sparsity achieved through regularization. In ‘sparse training,’ “sparse” is an adjective that directly modifies “training,” emphasizing the use of sparse methodologies during the training process. Due to changes in the manuscript after revision, the content originally on lines 170, 202, and 193 now appears on lines 200, 237, and 228, respectively. Please review these lines for the changes made.

Comments 8: In line 220, LBN1 in Eq. 4 is explained, whereas there is no LBN1 in Eq.4 in line 231. Maybe, replacing it with LCh1 is the appropriate correction.

Response 8: Thank you for your correction. We change ‘LBN1’ to ‘LCh1’ in line 220 of the original manuscript, and the modification is now reflected in line 255 of the new manuscript.

Round 2

Reviewer 1 Report

Comments and Suggestions for Authors

1)From Table 6, Custom1 is labelled the best ,Maybe It is not correct.  

‘Custom2’ and ‘Custom3’ respectively reduce the ratio of retained channels in layers with fewer and more important channels. The two have similar inference speeds, but the mAP differs by 0.67%.(Is it the number:96.77%-96.12%)?

2)Can the device recognize the same class rapeseed? I mean when the image includes two or five different rapeseeds, Is it Ok to give the correct detection?

3) Only five different types of rapeseeds?

4) Some table title or figure title is too long expressed. 

Comments on the Quality of English Language

The language should be dense.

Author Response

Comments 1: From Table 6, Custom1 is labelled the best ,Maybe It is not correct.  

‘Custom2’ and ‘Custom3’ respectively reduce the ratio of retained channels in layers with fewer and more important channels. The two have similar inference speeds, but the mAP differs by 0.67%.(Is it the number:96.77%-96.12%)?

Response 1: After careful consideration,we confirm that ‘Custom1’ is the optimal choice, as it has the highest mAP and a reasonably fast inference speed. And thank you for pointing out the error in the manuscript. The difference between ‘Custom2’ and ‘Custom3’ is 0.65%, and we make the corresponding correction in the manuscript at line 389.

Comments 2: Can the device recognize the same class rapeseed? I mean when the image includes two or five different rapeseeds, Is it Ok to give the correct detection?

Response 2: Thank you for your question. When an image includes multiple rapeseed varieties, our model is capable of detecting each variety. This capability is attributed to the implementation of the Mosaic data augmentation strategy, which randomly merges four images from the training set into one new image. This approach effectively simulates scenarios where a single image includes multiple rapeseed varieties.

Comments 3: Only five different types of rapeseeds?

 Response 3: Thank you for your question. During our experiment, we only employ five common varieties of rapeseed seeds, but their appearance features are typical and indicative of the general morphology of Brassica napus seeds. Our model has excellent adaptability, so we are confident that it will retain high efficiency when seeds from other rapeseed varieties are included in the dataset.

Comments 4: Some table title or figure title is too long expressed. 

Response 4: Thank you for your advice. We make the following changes to the figure titles:

Table 4: Performance comparison of our method and random branch selection method.

Figure 7: Scaling factor distributions in YOLOv7 (MB).

Figure 8: Influence of different pruning ratios on mAP.

Figure 9: Channel retention ratio of each layer.

Thank you again for your valuable time and professional advice.